# The crystal structure of a simian Foamy Virus receptor binding domain provides clues about entry into host cells

Ignacio Fernández[1], Lasse Toftdal Dynesen [2], Youna Coquin[2], Riccardo Pederzoli[1], Delphine Brun[1], Ahmed Haouz [3], Antoine Gessain[2], Félix A. Rey [1], Florence Buseyne [2] & Marija Backovic [1] ✉

The surface envelope glycoprotein (Env) of all retroviruses mediates virus binding to cells and fusion of the viral and cellular membranes. A structure-function relationship for the HIV Env that belongs to the Orthoretrovirus subfamily has been well established. Structural information is however largely missing for the Env of Foamy viruses (FVs), the second retroviral subfamily. In this work we present the X-ray structure of the receptor binding domain (RBD) of a simian FV Env at 2.57 Å resolution, revealing two subdomains and an unprecedented fold. We have generated a model for the organization of the RBDs within the trimeric Env, which indicates that the upper subdomains form a cage-like structure at the apex of the Env, and identified residues K342, R343, R359 and R369 in the lower subdomain as key players for the interaction of the RBD and viral particles with heparan sulfate.

Spumaretroviruses, also known as Foamy viruses (FVs) are ancient retroviruses that have co-evolved with vertebrate hosts for over 400 million years[1,2]. FVs are prevalent in non-human primates, which can transmit them to humans, most often through bites[3]. Unlike their better-studied *Orthoretrovirinae* relatives (HIV being the most notable member) FVs have extremely slowly mutating genomes and do not induce severe pathologies despite integrating into the host genome and establishing lifelong persistent infections[4,5]. These features, along with broad tropism and host range[6], make FVs attractive vector candidates for gene therapy[7].

Viral fusion proteins drive membrane fusion through a conformational change, which can be triggered by acidification in an endosomal compartment and/or binding to a specific cellular receptor[8,9]. FVs enter cells by endocytosis, with fusion of the viral and cellular membranes occurring in the endosomal compartment in a pH-sensitive manner, leading to nucleocapsid release into the cytosol[10]. The exception is the prototype FV (PFV), which can also fuse at the plasma membrane[11]. The FV envelope (Env) glycoprotein belongs to a class I fusogens[12], which are synthesized as single-chain precursors that

fold into trimers, and whose protomers are subsequently cleaved in the Golgi compartment during transport to the cell surface. FV Env is cleaved at two sites by cellular furin during maturation, giving rise to 3 fragments: the leader peptide (LP), the surface (SU) subunit, which includes the receptor-binding domain (RBD), and the transmembrane subunit (TM), which harbors the fusion machinery. The structural information available for FV Env is limited to cryo-electron tomography (ET) of viral particles and a 9 Å cryo-electron microscopy (EM) reconstruction of PFV Env[13], which revealed LP-SU-TM trimers arranged in interlocked hexagonal assemblies[13,14], with an architecture that is different to that of HIV Env trimers[15].

Heparan sulfate (HS) is an attachment factor for PFV and feline FV[16,17] but the requirements for a surface or intracellular receptor, which would trigger membrane fusion by FV Env, remain unclear. The search for a receptor has been complicated by FV binding to HS, which is ubiquitously expressed on cells, masking potential entry receptor candidates. A bipartite RBD, consisting of two discontinuous regions of the polypeptide chain, was identified by screening a panel of recombinant SU truncations for binding to cells[18]. The RBD was also

[1]Institut Pasteur, Université Paris Cité, CNRS UMR3569, Unité de Virologie Structurale, 75015 Paris, France. [2]Institut Pasteur, Université Paris Cité, CNRS UMR3569, Unité d'Épidémiologie et Physiopathologie des Virus Oncogènes, 75015 Paris, France. [3]Institut Pasteur, Université Paris Cité, Plateforme de cristallographie-C2RT, CNRS UMR 3528, 75015 Paris, France. ✉e-mail: marija@pasteur.fr

shown to be the main target of neutralizing antibodies in infected humans[19,20].

FV Env is heavily glycosylated, with at least 13 predicted N-linked glycosylation sites. Mutational analyses have revealed that three of these N-sites are essential for PFV infectivity−2 located in the TM subunit, and one in the RBD. The latter site, referred to as the glycosylation site 8 or N8[21] is conserved across the FV subfamily, and has been suggested to play a direct role in binding to a receptor[18] (to distinguish the nomenclature of the predicted N-linked glycosylation sites (N1 to N15) from the single letter symbol for asparagine (N), the former will be underlined throughout the text). The remaining molecular determinants of the RBD interaction with host cells remain elusive, largely due to a lack of structural information, which has precluded rational approaches to mutagenesis and functional analyses. A high-resolution structure of the FV RBD, structural information regarding the RBDs organization within the Env trimer and how the RBDs contribute to the Env activation are not available.

In this manuscript we present the X-ray structure of the RBD from a zoonotic gorilla FV at 2.57 Å resolution, which reveals a novel fold. By rigid-body docking into the available cryo-ET reconstruction of a PFV Env trimer[13], we derived a model for the RBD organization in the Env, and identified residues involved in HS binding, with functional and evolutionary implications discussed. Structural knowledge on the FV RBD is critical for understanding virus-cell interactions, the initial step that triggers Env to mediate membrane fusion, and for insights into the antibody-mediated neutralization by human hosts. Our data provide a framework for rational structure-guided mutagenesis studies necessary for discerning the molecular basis of different steps of FV entry into host cells.

## Results

### The X-ray structure of the SFV RBD reveals a novel fold

Recombinant RBDs from several simian FV (SFV) strains were tested for production in *Drosophila* S2 insect cells, and only the RBD from gorilla SFV (strain SFVggo_huBAK74[22], genotype II; abbreviated as 'GII' herein) was expressed in high enough yields and formed crystals. The RBD was expected to be heavily glycosylated due to 8 predicted N-glycosylation sites (Fig. 1a). To increase the chances of generating well-diffracting crystals, a fraction of the purified protein was enzymatically deglycosylated (RBD^D) with endoglycosidases H and D, as described in the Methods section. Crystals were obtained for the RBD^D, as well as for the untreated protein (RBD^G). The RBD^D diffracted better (2.57 Å) than RBD^G (2.80 Å) and the structural analyses presented below were carried out using the RBD^D structure, unless otherwise noted. The data collection and structure determination statistics for both crystal forms are summarized in Table S1.

The SFV RBD folds into two subdomains, each with α + β topology, which we refer to as 'lower' (residues 218–245, 311–369, and 491–524) and 'upper' subdomains (residues 246–310 and 370–490) in reference to their positioning with respect to the viral membrane[13] (see below). The RBD has a bi-lobed, bean-like shape with the longest dimension of ~65 Å. The upper subdomain forms a wider (~45 Å diameter) and the N- and C-termini a narrow lobe (~20 Å diameter), which is also termed lower subdomain (Fig. 1b). The lower subdomain is comprised of a three-helix bundle (α1, α7, α8) that packs against an antiparallel, twisted four-stranded β-sheet (β14-β1-β5-β15) and against helix α2 (residues 333-346) that lays perpendicularly on the side of the bundle. Within the helical bundle and the β-sheet, the regions proximal to the N- and C-termini are tied together by disulfide bonds (DS) DS1 (C228-C503) and DS2 (C235-C318), respectively. The upper subdomain is formed by two long excursions out of the lower subdomain: ~70 residues connecting strands β1 and β5, and ~130 residues connecting η4 and β14 (Fig. 2). The polypeptide chain thus extends upwards and back twice, forming at the end the outer strands (β14 and β15) of the lower subdomain β-sheet. These secondary structural elements in the lower

subdomain encompass a prominent hydrophobic core that extends into the upper subdomain, which has lower secondary structure content (Table S2) and is stabilized by several networks of polar interactions (Fig. S1 and Table S3). Four notable protrusions, designated loops 1 to 4 (L1-L4) emanate from the upper domain: loop 1 (L1, residues 253–270, connecting β2 and η1), loop 2 (L2, residues 276–281, connecting η1 and β3), loop 3 (L3, residues 414–436, connecting α5 and β9) and loop 4 (L4, residues 446–453, connecting β10 and β11). Loops L3 and L4 are particularly mobile in our structures as indicated by high B-factors (>105 Å²) for their Cα atoms (Fig. S2). Electron density was observed for the 8 predicted N-glycosylation sites (Fig. 1c), allowing the modeling of at least one N-acetyl glucosamine (NAG) at each site (Fig. 2 and Fig. S3a).

Searching the PDB databank using the DALI algorithm[23], with the RBD and its substructures as queries, did not yield any meaningful results. Comparative analyses with the available structures of the RBDs from Orthoretroviruses (Fig. S4) did not reveal structural similarity, either at the level of the secondary structure topology or the three-dimensional fold. Therefore, the SFV RBD represents, to the best of our knowledge, an unprecedented fold.

### The sugar attached to the strictly conserved 8th N-glycosylation site (N8) plays a structural role

There were no major differences between the X-ray structures of RBD^D and RBD^G (their superposition yielded a root mean square deviation (rmsd) below 1 Å (Fig. S3b, c), except for the different number of sugar units that we could build into the electron density maps (Fig. S3a). A prominent feature of the upper subdomain is the eighth N-linked sugar (N8)[18] attached to the α4 helix residue N390. The N390 side chain and the first two attached NAG residues are buried in the RBD, rendering the EndoH/D cleavage site inaccessible (Fig. 3b), which allowed building 10 sugar residues in RBD^G and 8 in the deglycosylated protein crystals (Figs. 2 and 3). The N8 glycan emerges from a cavity that has N390 at its base and extends upwards, remaining in contact with the protein and preserving the same conformation in both crystal forms. Structural analyses revealed that the glycan establishes extensive van der Waal contacts with the residues underneath (buried surface area of 803 Å²) and forms hydrogen bonds with main-chain atoms from Y394 and I484 and the side chain of E361 (Fig. S5). The glycan moiety covers a well-conserved and hydrophobic surface (Fig. 3a, b) and thus maintains the RBD fold and prevents aggregation, consistent with the reported misfolding and low levels of the secreted PFV SU with a mutation in the N8 site[18]. N8 is the only N-glycosylation site in SU that is strictly conserved across the FV subfamily (Fig. S6), and the hydrophobic patch residues laying beneath it are conserved as well (Fig. 3c). Thus, N8 likely plays an important structural role in all FV RBDs.

### The RBD fold is predicted to be conserved across the *Spumaretrovirinae* subfamily

To investigate potential conformational differences between RBDs from different species, we used AlphaFold 2 (AF2)[24] software for ab initio prediction of the RBD structures from members of each of the 5 FV genera, some of which exist as two genotypes due to the modular nature of FV Env[25]. Within each FV Env, a ~250-residue long region within the RBD, termed the variable or 'SU^var', defines two co-circulating genotypes, I and II, which have been found in gorillas[26], chimpanzees[19] and mandrills[25], among others. The SU^var regions share less than 70% amino acid sequence identity (Fig. S6), while the rest of Env residues are highly conserved (>95% sequence identity). The SU^var is located within the upper subdomain and encompasses loops L1-L4 (residues 282-487 in GII RBD; Figs. S6 and S7).

All the generated AF2 models have high-confidence metrics (Fig. S8) and display a conserved fold in agreement with an amino acid sequence identity >30%. Significant deviations were found only in the loops within the SU^var. The Template Modeling score (TM-score), which

is, unlike the rmsd, a length-independent measure of structural similarity[27] has the average value of 0.89 for the 11 compared structures. The AF2 model of the GII RBD and our experimentally determined structure of the same strain superimpose with a TM-score of 0.96 and rmsd of 1.5 Å for 320 out of 328 Cα atoms aligned, confirming the high accuracy of the AF2 model. A 'common core' (CC), which includes the ensemble of residues with Cα rmsd values smaller than 4 Å for all the pairwise superpositions, was calculated by the mTM-align webserver[28]. The CC of the FV RBD contains 239 out of 308 aligned residues (Fig. S9a), with most CC residues belonging to the secondary structure elements forming the lower subdomain. The loops in the upper subdomain are largely not a part of the CC (Fig. S9).

## Fitting of the RBD atomic model into Env cryo-EM density maps reveals the RBD arrangement in the Env trimer

To investigate the RBD arrangement within trimeric Env, we fitted the RBD atomic model into the 9 Å cryo-EM map reported for trimeric PFV (a chimpanzee genotype I FV) Env expressed on Foamy viral vector (FVV) particles[13]. The fitting was justified by the high structural conservation between gorilla and chimpanzee RBDs, indicated by a TM-score of 0.88 for the superposition of the GII RBD structure and the predicted PFV RBD model (Fig. S8).

The RBD fitting was performed with the fit-in-map function in Chimera suite[29] (Fig. 4a) as described in material and methods. The correlation coefficient of 0.96 strongly suggests that the recombinantly

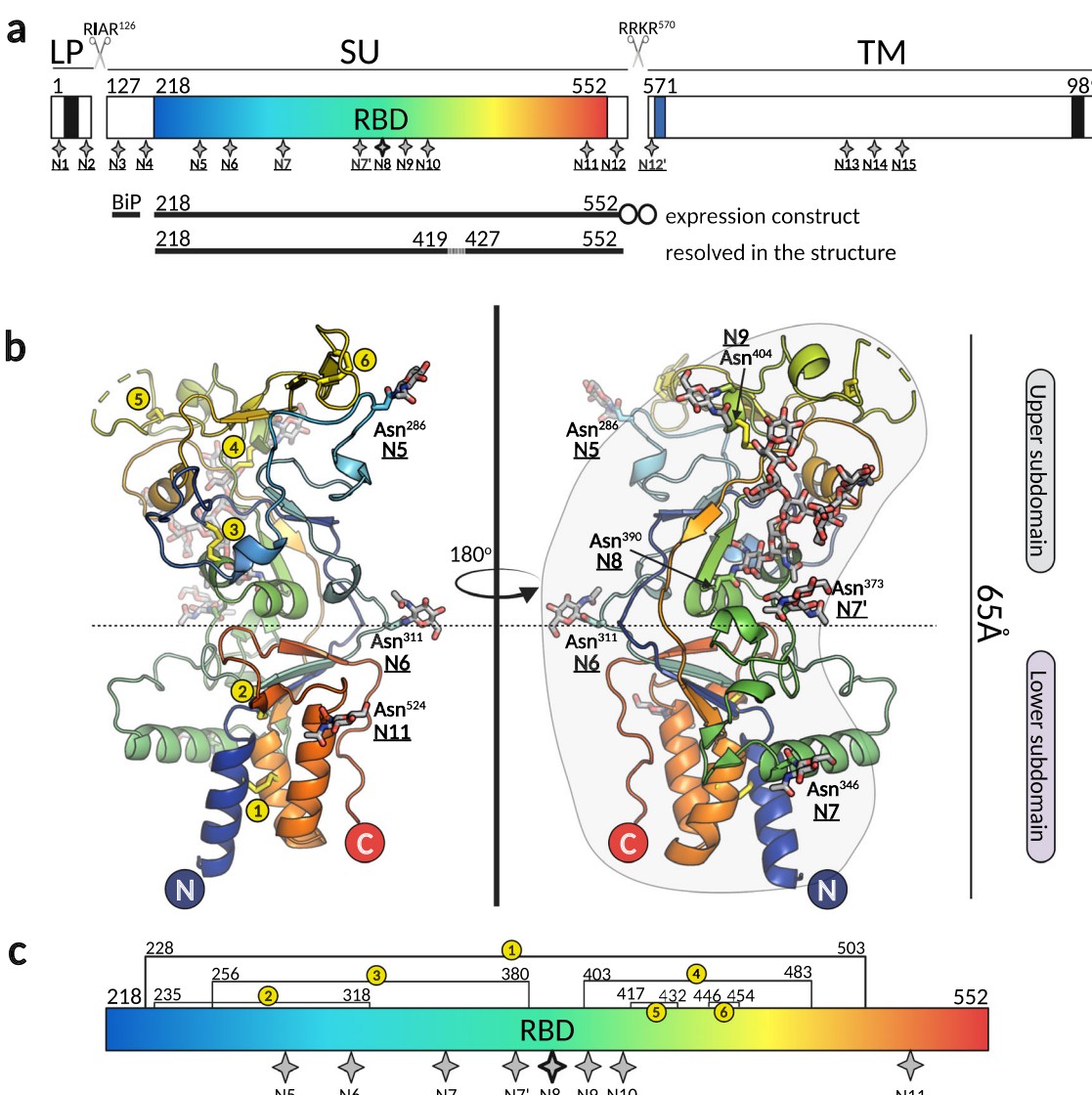

**Fig. 1 | Overview of the novel fold adopted by the SFV RBD. a** Schematic representation of SFV Env protein organization indicating the three constituent chains: leader peptide (LP), surface subunit (SU), and transmembrane subunit (TM). The transmembrane domains anchoring the LP and TM in the membrane are represented as black boxes; the receptor-binding domain (RBD) within SU is highlighted in blue-red spectrum; the fusion peptide at the N-terminus of the TM is shown in blue. The furin sites between the LP and SU (RIAR[126]), and SU and TM (RRKR[570]) are indicated with scissors icons. The RBD expression construct contained the exogenous BiP signal at the N-terminus, residues 218 to 552 of the SFV gorilla GII Env and a double strep tag at the C-terminus (shown as two circles). The region comprising residues 420–426 is drawn as a dashed line because it was not seen in the electron density map. The 17 putative N-glycosylation sites for gorilla

SFV Env are indicated with star symbols and labeled N1 to N15, following the previously established nomenclature[22]. **b** The X-ray structure of the RBD[D] is shown in ribbon model colored from N- to C-terminus in blue to red spectrum, respectively. The dashed-line indicates the separation between the upper and lower subdomains. The N-glycosylation sites are indicated with N, and the sugars and the asparagine side chains carrying them are displayed as sticks. The figure was created with Pymol[65]. **c** Linear representation of the RBD. The 8 N-glycosylation sites with sugars built into the electron density are shown as gray stars, including site N10 (N411) that showed density for an attached carbohydrate in RBD[G] (molecule B), but not in RBD[D]. Site N8 (N390), which contains a long, partially buried sugar moiety is highlighted with a thicker outline. The locations of six disulfides are indicated with numbered yellow circles as in panel b. The figure was created in BioRender.com.

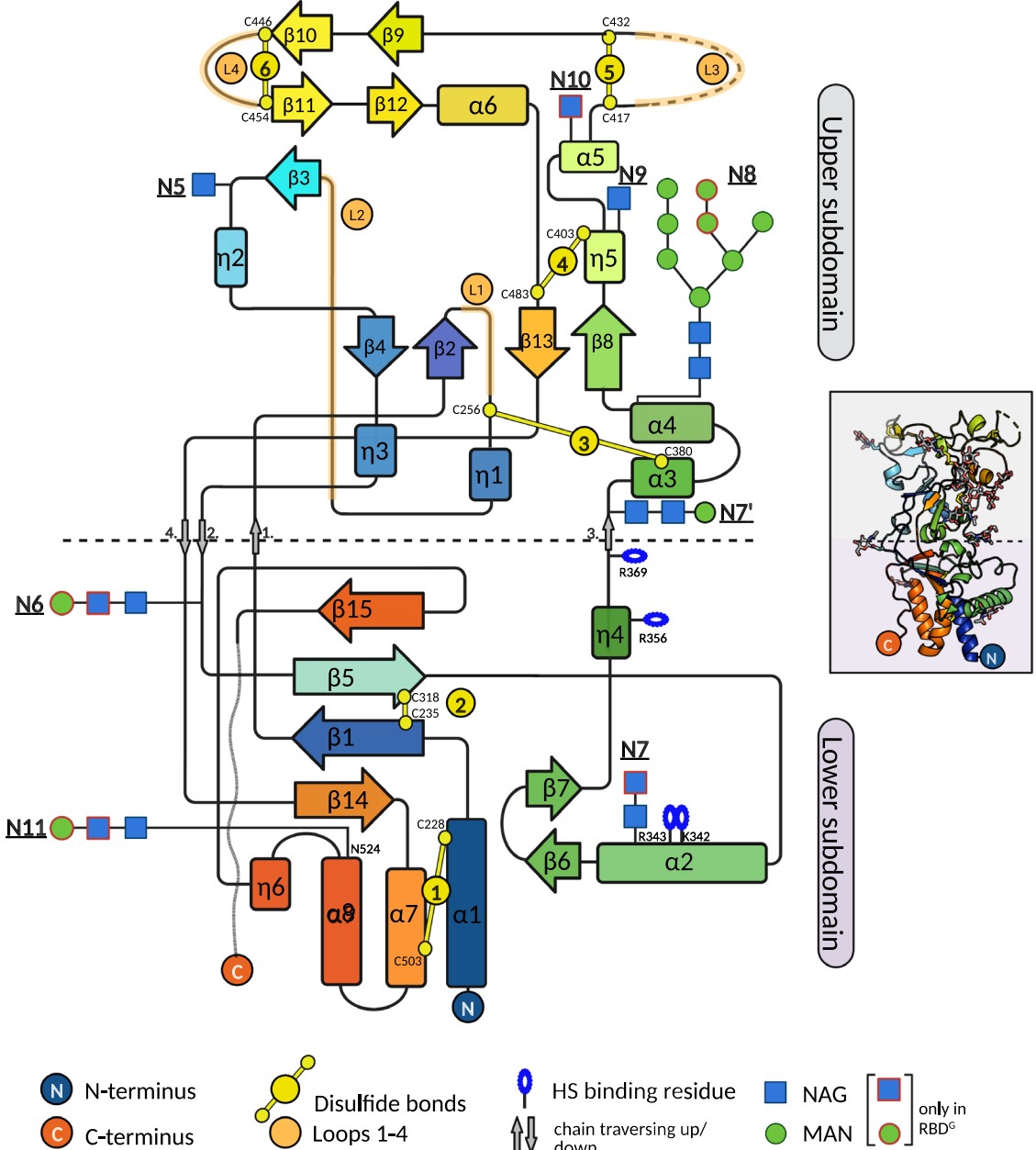

**Fig. 2 | SFV RBD secondary structure topology diagram.** The horizontal dashed-line designates the boundary between the lower and upper subdomains. The NAG and MAN units built only into the RBD$^G$ (and not RBD$^D$) structure are indicated with red frames. The figure was created with BioRender.com.

expressed RBD represents its biologically relevant conformation as observed at the surface of virus particles. The 7 N-linked glycans that we could resolve in the RBD$^D$ structure are all fully solvent-exposed (Fig. 4b), and additional density was observed in the PFV EM map for the sugars attached to N5, N6, N7, N8, N9 and N11, validating the RBD placements (the N7' site is absent from PFV Env, and no extra density was observed at this site). The RBD N- and C-termini point towards the membrane, indicating that the lower half of the Env density is occupied by the TM subunit and the remaining SU residues, as previously suggested[13].

The three fitted RBDs are arranged around a central cavity at the apex (membrane-distal region) of Env (Fig. 4a). The analyses of the macromolecular surfaces of the trimeric RBD model, carried out in PDBePISA[30], revealed a limited interprotomer interface (<10% of the entire RBD solvent accessible surface) established by loops L1-L4 that form a ring-like structure at the RBD apex, leaving most of the RBD exposed (Fig. 4b). According to the model, the three L1 loops likely

engage in homotypic interactions at the center of the RBD, forming an inner ring, while each L3 loop contacts L4 and L2 of a neighboring protomer. The residues at the interface of the docked models, which could potentially form non-covalent contacts (Fig. S10b, c) and the length of the loop regions (Figs. S9a and S10a) are poorly conserved across the FV family. It is important to note that PFV Env TM subunits trimerize forming a prominent central coiled coil[13], a hallmark of all class I fusion proteins. Thus, the potential interfaces established between the RBDs would be only one of the contributing Env trimerization sites.

To assess the importance of the loops for Env function, we generated FVVs carrying GII Env with deletions of loops L2 and L4 (ΔL2 and ΔL4, respectively). The amount of secreted mutant FVV particles was reduced over 50-fold (Fig. S11a) and their binding to cells was decreased 2 to 4-fold compared to the FVVs with WT Env (Fig. S11a). The infectivity of both mutants was however below the detection limit of our assay (Fig. S11c, d) indicating that despite poor sequence

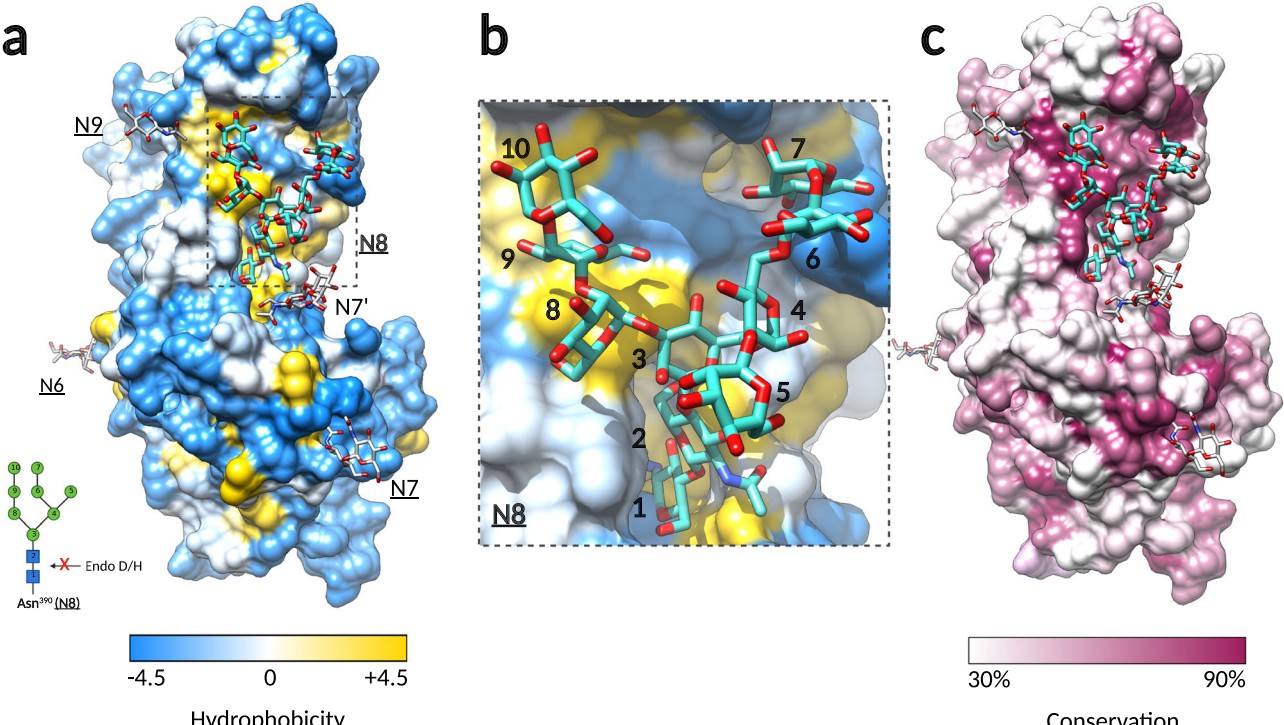

**Fig. 3 | The oligosaccharide linked to N390 plays a structural role in the RBD. a** Molecular surface representation of the SFV RBD colored by residue hydrophobicity. Hydrophobicity for each residue was calculated according to the Kyte and Doolittle scale[66] in Chimera[29], with the gradient color key indicating the lowest hydrophobicity in blue, to the highest hydrophobicity in yellow. The sugars at sites N6, N7, N7', and N9 are displayed as white sticks, and the sugar attached to N8 as cyan sticks. The inlet shows the bond cleaved by glycosidases Endo D/H, which is protected in N8. **b** The N8 sugar attached to N390 covers a hydrophobic region. Magnified region within the dashed-line rectangle in panel a Is shown. The NAG and MAN residues are labeled with numbers that correspond to the N-oligosaccharide drawn in the inlet of panel a. **c** The hydrophobic patch covered by N8 is well-conserved. The SFV RBD surface is rendered by residue conservation in Chimera[29], according to the % of the identical residues in the 11 FV Env sequences (alignment shown in Fig. S6). Residues conserved in less than 30% and more than 90% of sequences are colored in white and purple, respectively, and residues in between with a white-purple gradient, as indicated on the color key below the surface representation.

conservation, loops L2 and L4 play a relevant structural and/or functional role for the activity of Env.

## Positively charged residues in the lower subdomain form a heparan sulfate (HS)-binding site

To locate potential HS-binding regions in the RBD, we investigated the electrostatic potential surface distribution and identified a large, continuous surface patch in the lower subdomain with a strong positive potential (Fig. 5a). We next analyzed the RBD structure with the ClusPro server that predicts putative HS-binding sites by docking a HS tetrasaccharide onto the protein surface[31]. K342 and R343 in helix α2, R359 in the proceeding helix η4, and R369 in an extended chain region were among the residues that had the highest number of contacts with the HS models that were docked onto the surface (Fig. 5b, c). The four residues also mapped within the positively charged region in the lower subdomain.

Based on the ClusPro predictions, we produced two GII RBD variants (K342/R343, termed 'mut1', and R359/R369, termed 'mut2') and tested their binding to HS immobilized on a Sepharose matrix (Fig. 6a). The two RBD variants eluted at the same volume on size exclusion chromatography consistent with the expected size of a monomer (Fig. S12), indicating that the introduced mutations did not cause protein misfolding. The WT RBD was retained on the heparin column and eluted at 300 mM sodium chloride concentration, while the mut1 and mut2 variants were not retained and eluted in the flow-through fraction. The observed loss of heparin-binding capacity strongly suggests that residues K342, K343, R359, and R369 are directly involved in interactions with HS.

We used flow-cytometry to investigate the interaction between the GII RBD and HS on cells (Fig. S13a). We found that the monomeric RBD did not bind to HT1080 cells even at high protein concentrations (Fig. 6b). We therefore tested a longer construct, the GII Env ectodomain, which spontaneously forms trimers, hypothesizing that an oligomer would yield higher signal due to avidity effects. The trimeric ectodomain bound to HT1080 cells (Fig. 6b), so the K342A/R343A and R356A/R369A mutations (mut1 and mut2, respectively) were introduced into the ectodomain background to render them suitable for flow-cytometry experiments. We compared the binding of the Env ectodomains to HT1080 and to BHK-21 cells (Fig. 6c), which are both susceptible to infection by gorilla FVs. We quantified HS expression levels by flow-cytometry concomitantly with the binding experiments and verified that BHK-21 cells expressed lower HS levels than HT1080 cells, as had been reported[17] (Fig. S13c). The HS expression levels were 10 to 30-fold lower on BHK-21 cells compared to HT1080 cells (Fig. 6c) and the binding of the WT ectodomain to BHK-21 cells was lower than to HT1080 cells at the highest protein concentrations tested. The binding signal was dose-dependent and one log lower for mut1 and mut2 ectodomain variants in comparison with the WT protein on both cell lines (Fig. 6c).

To prove that the designed mutations specifically affected the interaction with cellular HS, we measured binding to HT1080 cells that were pre-treated with heparinase, which removed more than 90% of HS from the cells (Fig. S13d). Binding of the WT ectodomain to heparinase-treated cells was diminished about 100-fold when compared to buffer-treated cells, while the mut1 and mut2 variants did not bind, independently of heparinase treatment (Fig. 6d).

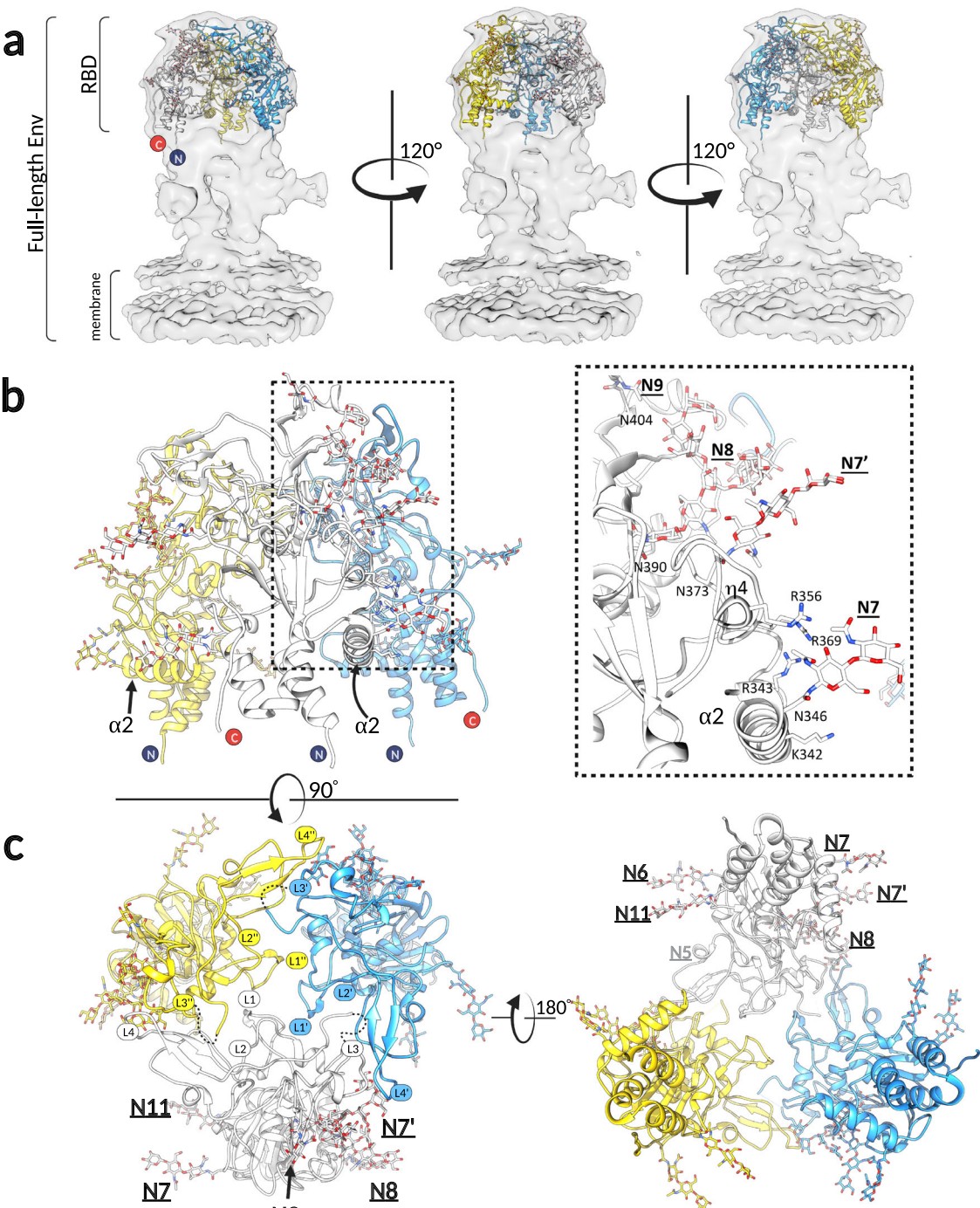

**Fig. 4 | The RBDs form a trimeric assembly at the apex of the full-length Env.**
**a** Three SFV RBD[D] protomers were fitted in the 9 Å cryo-EM map (EMBD: 4013) obtained by cryo-EM 3D reconstruction of the full-length PFV Env expressed on viral vector particles[13]. The map is shown in light gray surface, and RBDs in cartoon mode, with each protomer colored differently (yellow, white, light blue). **b** The three RBDs, fitted as explained in panel a, are shown to illustrate that the α2 and η4 helices, which carry the HS-binding residues (K342, R343, R359, R369), and the N-linked glycosylations (N6, N7, N7', N8, N9 and N11) point outward and are solvent accessible. The boxed region on the left panel is magnified and displayed on the right panel (only one protomer, colored in white, is represented for clarity purposes). **c** The views at the trimeric RBD arrangement from the top i.e. looking at the membrane (left) and bottom i.e. looking from the membrane (right) are shown. The RBDs form interprotomer contacts via the L1-L4 in the upper domain. The loops belonging to each protomer are designated as L, L', and L". Images were generated in Chimera[29].

The importance of residues K342, R343, R356, R369 for virus binding to HS was tested using FVVs that express either WT, mut1 or mut2 Env on their surface. The total number of FVV particles released by the transfected cells, measured by RT-qPCR, was ∼6-fold lower for mut1 compared to WT FVVs, while mut2 had the same particle production as WT (Fig. S14a). The infectious titers were 34- and 65-fold lower for mut1 and mut2, respectively, compared to WT (Fig. S14b). The proportion of infectious particles, the infectious titer (Fig. S14b), divided by the total number of FVVs (Fig. S14a) was 0.7% for WT Env FVVs, while the values for mut1 and mut2 FVVs were 3- and 22-fold lower, respectively (Fig. 6e). We measured the binding of FVVs to cells by RT-qPCR and found that the binding was also reduced 3- and 23-fold

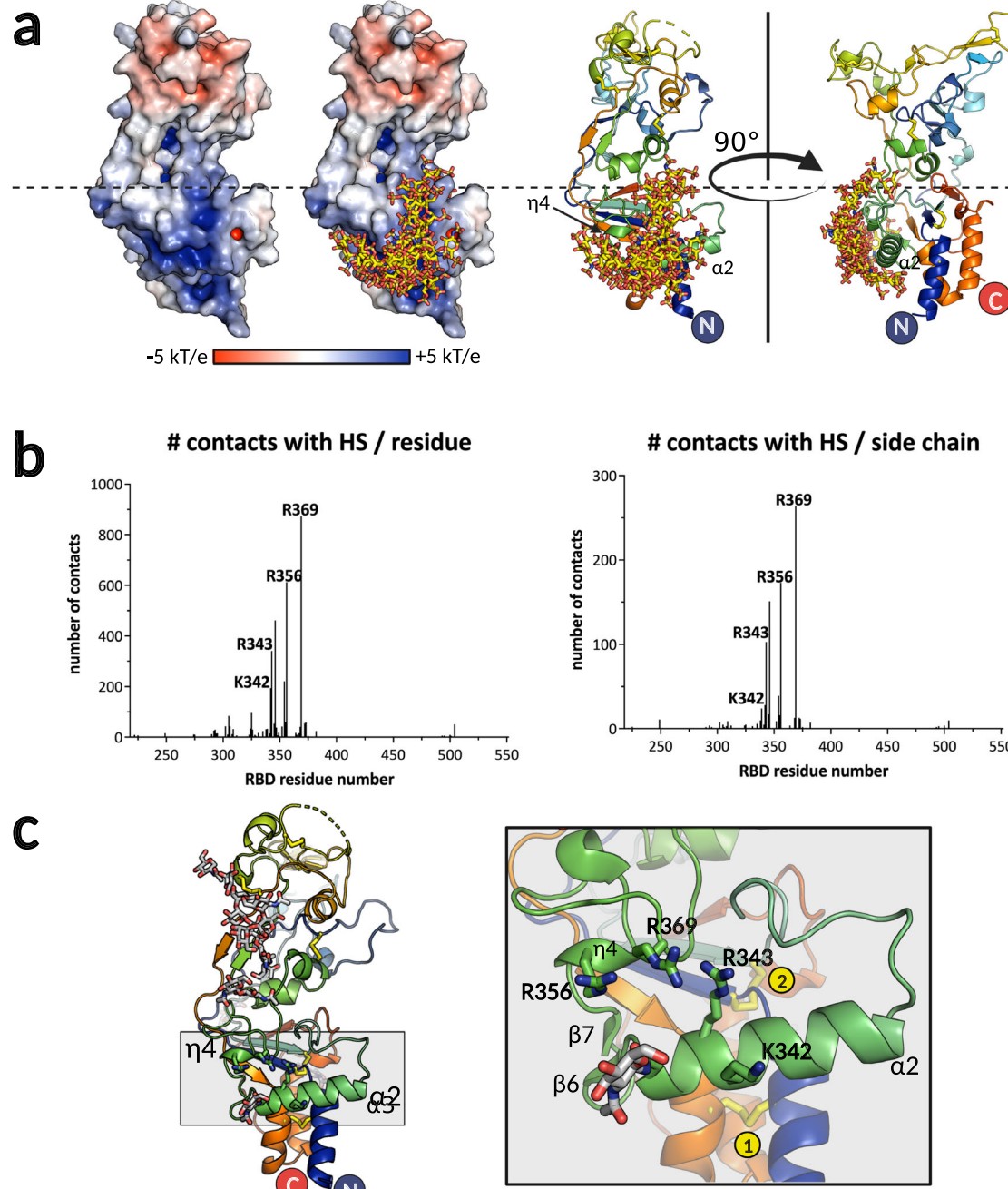

**Fig. 5 | Prediction of HS-binding residues and design of the variants impaired in binding. a** Electrostatic potential distribution was calculated using Adaptive Poisson-Boltzmann Solver module[67] in Pymol[65] and plotted on the solvent excluded surface of the RBD, with red corresponding to the negative, and blue to positive potentials (two left panels). The ensemble of HS molecules modeled by ClusPro[31] map to the lower subdomain and are displayed in sticks on the two right panels. The RBD is shown in cartoon model and in two orientations to illustrate the location of predicted HS-binding secondary structure elements. **b** Predicted number of contacts per residue and per side chain atoms calculated by ClusPro and plotted for each RBD residue, revealing the most likely candidates to be engaged in HS binding. Source data are provided as a Source Data file. **c** Structure of RBD is shown in cartoon, with the region containing α2 and η4 helices highlighted in gray. Magnification of the gray boxed region is shown on the right panel, with the relevant secondary structure elements and predicted HS-binding residues shown in sticks. Two disulfide bonds are indicated with yellow circles. The figure was created with Pymol[65] and BioRender.com.

for FVVs carrying mut1 and mut2 Envs, respectively, compared to the WT Env FVVs (Fig. 6f). Thus, binding to cells and entry levels were decreased to the same extent for the FVVs carrying Env proteins with mutations in the HS-binding site.

The results described for the recombinant Env proteins (Fig. 6c, d) and FVVs carrying full-length Env (Fig. 6f) agree with the biochemical data (Fig. 6a) and demonstrate that residues K342, R343, R356, R369 play a crucial role in virus interaction with HS.

## Discussion

### FV RBD adopts a novel fold and is composed of two subdomains

We determined the X-ray structure of the RBD from a gorilla FV, revealing a three-dimensional fold (Figs. 1 and 2) distinct from the available Orthoretrovirus RBD structures i.e., RBD from the Friend murine leukemia virus, feline leukemia virus, human endogenous retrovirus EnvP(b)1 (gammaretrovirus genus)[32–34], and gp120 from HIV[35] (lentivirus genus) (Fig. S4). This finding expands the repertoire of

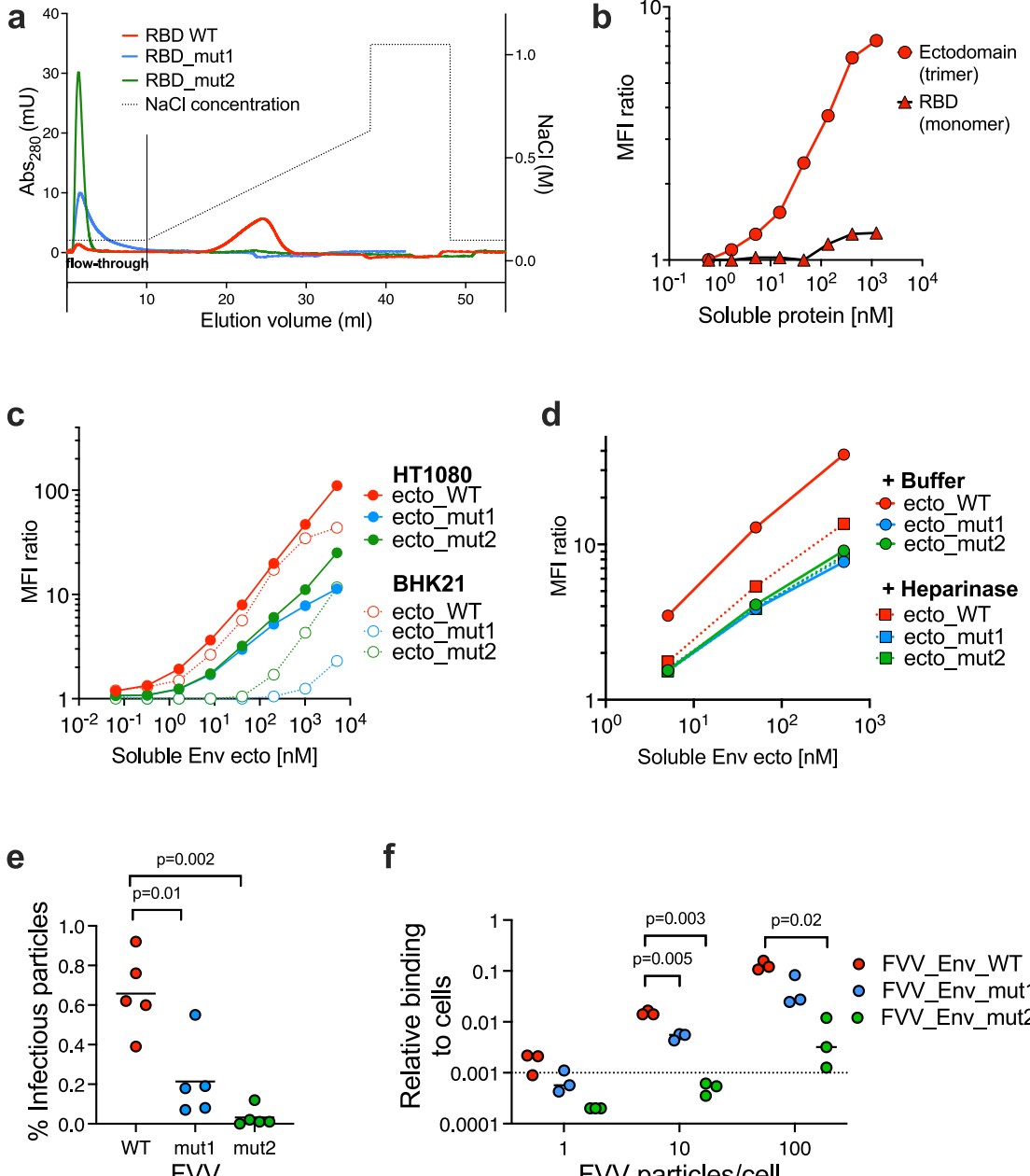

**Fig. 6 | The SFV RBD residues K342, R343, R356 and R369 mediate Env binding to HS. a** Heparin-Sepharose chromatogram of the recombinant SFV RBD, WT (red) and variants with mutations in HS-binding residues, mut1 (K342A/R343A) in blue, and mut2 (R356A/R369A) in green. Dotted line indicates salt concentration, plotted on the right y-axis. Source data are provided as a Source Data file. **b** Binding of recombinant WT RBD and Env ectodomains to HT1080 cells. SFV RBD and ectodomain binding levels were expressed as the ratio of MFI from protein treated to untreated cells. To be comparable with the RBD, the concentration for the ectodomain is calculated and plotted for the monomeric protein. Gating strategy is presented in (Fig. S13a). Source data are provided as a Source Data file. **c** Binding of WT, mut1 (K342A/R343A), and mut2 (R356A/R369A) recombinant Env ectodomains to HT1080 and BHK-21 cells. SFV Env binding level to live single cells was expressed as the ratio of MFI from protein treated to untreated cells (Fig. S13b). Mean from two independent experiments are shown. Cell HS expression levels were monitored on the day of each of the two experiments (the HS staining levels were 85.4 and 61.6 for HT1080 cells, 8.40 and 2.68 for BHK-21 cells (Fig. S13c)). Source data are provided as a Source Data file. **d** Binding of WT, mut1 (K342A/R343A), and mut2 (R356A/R369A) recombinant Env ectodomains to HT1080 cells, treated with heparinase or buffer, was quantified at increasing ectodomain concentrations.

Ectodomain binding level to live single cells was expressed as the ratio of MFI from protein treated to untreated cells (Fig. S13b). Mean values from two independent experiments are shown. HS surface expression and removal were quantified by HS and ΔHS-specific antibodies (Fig. S13d). Source data are provided as a Source Data file. **e** Five batches of FVVs carrying WT (red), mut1 (blue) and mut2 (green) Envs were produced. Each batch is represented with a single dot; the black lines indicate the mean values. The percentage of infectious FVV particles carrying WT, mut1 or mut2 Env was calculated as the ratio between the number of infectious particles (determined by titration on susceptible cells, Fig. S14a) and the amount of vector particles obtained by RT-qPCR (Fig. S14b). The mutant FVVs were compared to the WT FVVs using the two-way paired t-test, with p-values indicated on the graph. Source data are provided as a Source Data file. **f** Binding of FVVs carrying the WT, mut1 or mut2 Env to HT1080 cells. Three batches of FVVs were incubated with HT1080 cells on ice for 1 h at different number of particles/cell ratios, before washing and quantifying the remaining vector particles by RT-qPCR. The dotted line represents the quantification threshold. The FVVs carrying the mutant and WT Envs were compared using the two-way paired t-test, with p-values indicated on the graph. Source data are provided as a Source Data file.

unique FV features (assembly, particle release[36], replication[37]) that are not shared with Orthoretroviruses, and is not surprising considering the lack of Env sequence conservation between the *Orthoretrovirinae* and *Spumaretrovirinae* subfamilies.

Structural information is available for the RBDs of some Orthoretroviruses. In the case of the gammaretroviruses, the RBD is relatively small (~200 residues) and folds into an antiparallel β-sandwich with two extended loops that give rise to a helical subdomain that sits on top of the β−sandwich[33] (Fig. S4). The helical subdomain defines the tropism for cellular receptors[38] and shows high sequence variability within the genus. In contrast, lentiviruses such as HIV have a receptor-binding region that is larger (~450 residues), encompassing most of the SU subunit termed gp120. HIV interacts with its cognate receptor CD4 through gp120, which is folded into two subdomains, inner and outer, with the receptor-binding surface formed by secondary structure elements from both subdomains[35]. Variable loops projecting out from the gp120 core, participate in receptor binding, immune evasion[39], and are key players in the Env conformational dynamics. This Env 'breathing' entails different arrangements of the gp120 subunits and the loops: the closed (in which the loops form contacts), relaxed (with a larger degree of openness), and open (which is achieved after CD4 receptor binding)[40,41]. By comparing the RBD of Orthoretroviruses with that of FVs, it is possible to argue that the FV RBD global organization into two subdomains−the lower, which is better conserved, and upper, which contains the protruding loops and is variable in sequence−is reminiscent of the characteristics described above for the RBDs of orthoretroviruses. Whether the presence of similar features in HIV and FV Env SUs implies similar function of the loops in receptor binding and conformational flexibility remains to be determined.

## The RBDs form a cage-like structure at the membrane-distal side of Env

We fitted the experimentally determined RBD structure into the low-resolution map (Fig. 4a) obtained by cryo-EM single particle reconstruction of trimeric PFV Env expressed on FVV particles[13]. The resulting model of the RBD trimeric arrangement is consistent with the biochemical and functional data presented here i.e., as expected the HS-binding residues (K342, R343, R356, R369) and 7 N-linked carbohydrates map to the Env surfaces that are exposed to the solvent (Fig. 4c). According to our model, the L1-L4 loops, located at the top of the upper subdomain of each protomer, are in proximity to each other (Fig. 4), leaving, just below, a cavity that was clearly visible in the cryo-EM maps[13]. Based on these observations we have speculated that the interprotomer interactions participate in maintaining a pre-fusion Env conformation. We tested FVVs carrying Env variants with deletions in the loops L2 and L4 (Fig. S11) and showed that these changes modestly affected the FVV binding to cells, but resulted in the complete loss of infectivity, corroborating the possibility that the Envs with loop deletions may easily transition to the fusion-inactive, post-fusion conformation.

The loop sequences are poorly conserved across the FV family and have variable lengths (Figs. S9 and S10a). The superpositions of the AF2 models of 11 FV RBDs revealed slight structural differences, which were limited to the variable region containing the loops (Figs. S8 and S9). Poor conservation of the residues at the interface between RBDs in the trimeric Env is suggestive of a weak selective pressure and could indicate that the Env native state relies on different sets of interacting loop residues in different FVs. Alternatively, the RBD-RBD interface could involve polar interactions between main-chain atoms, although we would not expect them to be numerous, considering that only a small surface of the RBD is buried at the interface. The advantage of having loosely bound RBDs would in facilitating dissociation upon a fusion trigger delivered in the endosome (acidic pH) and/or by a specific cellular receptor. In that respect, the FV RBD loops could play a role equivalent to V1/V2/V3 loops in HIV Env, which provide

conformational flexibility[40,41]. It will be also important to discern the RBD molecular determinants, if any, that drive the membrane fusion at the plasma membrane, as used by PFV, in comparison to all the other FVs that fuse in the endosomes[11].

## The structure explains why the upper RBD domain can tolerate deletions and substitutions

Based on the ability of SU truncated variants to bind to cells, Duda et al. defined the RBD of PFV Env as a region spanning residues 225-555[18] (residues 226–552 in gorilla GII Env (Fig. S15a)). Within the proposed region, the central segment was found to be dispensable for cell-binding activity[18]. This segment (termed also RBDjoin[42]) encompasses loops L3 and L4, maps to the top of RBD and is clamped by two disulfide bonds (Fig. S5 and S15a). Its location, away from the HS-binding residues, is consistent with the ability of the PFV SU truncation lacking the equivalent region to bind to cells at the levels measured for WT protein[18]. The AF2 model of the PFV RBD lacking the RBDjoin region indeed reveals a 3D fold very similar to that of the complete RBD (Fig. S15b). In contrast, we show that the deletions of loops L2 and L4−in the context of trimeric GII Env−have a profound effect on infectivity, highlighting the different behavior of a soluble RBD, and RBDs within the full-length Env trimer.

## The lower RBD subdomain carries the residues involved in HS binding

Our data demonstrate that K342/R343 and R356/R369 are the key residues for the RBD interaction with HS immobilized on an inert matrix or expressed on cells (Fig. 6) and that HS is an attachment factor for gorilla FV, expanding upon previous reports for PFV[17]. The recombinant SFV GII RBD with the 4 mutations (K342A, R343A, R356A and R369A) had very low expression yields, but as expected did not bind to the heparin column. In SFV Envs, the residue at position equivalent to 343 in gorilla GII Env is always an arginine or lysine, while arginine is strictly conserved at position 356 (Fig. S6). Residues at positions 342 and 369 are less conserved among FV Envs, although they are usually surrounded by positive or polar residues. This suggests that the R343 and R356 may be important for HS binding in all FVs, while other positively charged residues, specific to each virus and located elsewhere within the patch with high positive electrostatic potential, could contribute to the HS binding in a virus-specific context (Fig. 5a).

Existence of an FV receptor had been proposed by Plochmann et al. since a total lack of HS did not abolish FV infection, although HS has also been proposed to function as a true FV receptor[16]. The residual Env binding to cells devoid of HS, which we observed both for the WT and the HS-binding impaired variants (Fig. 6d), is consistent with the presence of additional cell receptor(s) in FV entry. The HS-binding defective Env variants we generated will be useful tools in the search of potential proteinaceous receptors, as they eliminate binding to HS, which is a widely expressed attachment factor.

## Limitations of the study

The high correlation coefficient we obtained for fitting the GII RBD X-ray structure in the PFV Env EM map strongly suggests that the general location of the RBD loops in our trimeric model is valid. The contact residues and interactions they establish can however not be accurately inferred because we fitted the GII RBD crystal structure in a cryo-EM map obtained for a different virus (PFV) at a low resolution (9 Å), precluding the refinement of side chains. In addition, all the RBD loops have high B factors (Fig. S2) and 7 residues could not be built in L3. As AF2 assigned low pLTTD values to the residues in the loops of other FV RBDs, identification of the contact residues at their RBD interfaces was not possible either.

Our data indicate that the L2 and L4 are important for the Env activity in entry, but do not provide direct evidence for deducing the

importance of these regions for the stability and pre-fusion conformation of the Env. As we performed the fitting in the low-resolution map, we can also not exclude the possibility that—in the context of Env trimer—RBDs can adopt varying degrees of openness, similar to gp120 in HIV Env[43]. The definitive identification of the RBD interfaces within the Env and their role in the pre-fusion state conformation stabilization will require an atomic resolution structure of the full-length Env trimer. We tried predicting the structure of the trimeric full-length Env by AF2, but the attempt was not successful due to the large size of Env protomer (almost 1000 residues) and computational limitations of the server we have access to.

## Concluding remarks

In this manuscript we have described the first X-ray structure of a FV RBD and validated that the novel fold is the one adopted in native FV Env. We identified, within the RBD, two subdomains in terms of their structure, conservation, and function: the upper subdomain, which encompasses most of the genotype-specific region, and a more conserved, lower subdomain, important for binding to the attachment factor HS. We generated AF2 models for 10 additional FV RBDs, highlighting its conserved three-dimensional conformation. This information is critical for understanding virus-cell interactions and has provided a framework for structure-driven mutagenesis studies necessary for establishing the molecular basis of FV entry and recognition by neutralizing antibodies as described in Dynesen et al.[42]. The AlphaFold[24] algorithm cannot predict the arrangement of oligosaccharides at the surface of glycoproteins. The previously reported functional observations on FV Envs, along with the role of N8, can now be understood in light of the experimentally derived structure, underscoring the necessity for structure determination by experimental means. Identification of HS-binding residues will aid the search for additional putative FV receptor(s). Insights into the structure-function relationship of the metastable, multimeric, and heavily glycosylated FV Env, as well as unraveling the molecular basis of receptor activation and membrane fusion, will require integrated biology efforts and experimental structural methods.

## Methods

### Expression construct design (SFV RBD and ectodomains for HS-binding studies)

A flow-cytometry assay was developed by Duda et al. to detect binding of recombinantly expressed Foamy virus Env variants to cells[18]. By using a panel of SU truncations fused to the Fc region of murine IgG (immunoadhesins) the authors showed that the RBD—defined as the minimal region of the PFV Env sufficient for binding to cells—encompassed residues 225 to 555 (corresponding to residues 226 to 552 in gorilla FV RBD (GII-K74 strain, accession number JQ867464)[44] (Fig. S6)). When designing the expression construct for SFV RBD, we also considered the secondary prediction generated by the Phyre2 webserver[45]. Residue I225 was in the middle of a putative helix (residues 220–230), leading us to choose an upstream residue R218 as the N-terminus of the construct (Fig. 1a and Fig. S6).

The information on secondary structure predictions, obtained by Phyre2 webserver was also used to design the Env ectodomain construct, which starts after the first predicted transmembrane helix (S91) and encompasses residues up to I905.

### Recombinant SFV RBD and ectodomain production and purification

For structural studies, the RBD (residues 218–552, GII-K74 strain, Env accession number JQ867464) was cloned into a modified pMT/BiP insect cell expression plasmid (Invitrogen) designated pT350, which contains a divalent-cation inducible metallothionein promoter, the BiP signal peptide at the N-terminus (MKLCILLAVVAFVGLSLG), and a double strep tag (DST) (AGWSHPQFEKGGGSGGGSGGGSWSHPQFEK)

at the C-terminus[46]. This plasmid was co-transfected in *Drosophila* Schneider line 2 cells (S2) with the pCoPuro plasmid for puromycin selection[47]. The cell line has undergone selection in serum-free insect cell medium (HyClone, GE Healthcare) containing 7 µg/ml puromycin and 1% penicillin/streptomycin. For the protein production stage, the cells were grown in spinner flasks until the density reached -1 × 10$^7$ cells/ml, at which point the protein expression was induced with 4 µM CdCl$_2$. After 6 days, the cells were separated by centrifugation, and the supernatant was concentrated and used for affinity purification using a StrepTactin column (IBA). Approximately 20 milligrams of recombinant RBD were obtained per liter of S2 cell culture. The DST was removed by incubating the protein with 64 units of Enterokinase light chain (BioLabs) in 10 mM Tris, 100 mM NaCl, 2 mM CaCl$_2$, pH 8.0, at room temperature, overnight. The proteolysis reaction was buffer-exchanged into 10 mM Tris, 100 mM NaCl, pH 8.0, and subjected to another affinity purification, recovering the flow-through fraction containing the untagged RBD. The protein was concentrated and its enzymatic deglycosylation with EndoD and EndoH was set up at room-temperature following overnight incubation with 1000 units of each glycosidase in 50 mM Na-acetate, 200 mM NaCl, pH 5.5. The protein was further purified on a size exclusion chromatography (SEC) column Superdex 200 16/60 (Cytiva) in 10 mM Tris, 100 mM NaCl, pH 8.0, concentrated in VivaSpin concentrators to 8.2 mg/ml and used as such for crystallization trials.

For cell-binding experiments the RBD construct was cloned in a pcDNA3.1(+) derived plasmid, for expression in mammalian cells. The expression plasmid was modified by inserting a CMV exon-intron-exon sequence that increases the expression of recombinant proteins. The RBD was cloned downstream of the CD5 signal peptide (MPMGSLQPLATLYLLGMLVASCLG) with an enterokinase cleavage site and a DST tag in the C-terminus. The HS mutants were generated by site-directed mutagenesis. The plasmids coding for the recombinant proteins were transiently transfected in Expi293F™ cells (Thermo Fischer) using FectroPRO® DNA transfection reagent (Polyplus), according to the manufacturer's instructions. The cells were incubated at 37 °C for 5 days after which the cultures were centrifuged. The protein was purified from the supernatants by affinity chromatography using a StrepTactin column (IBA), followed by SEC on a Superdex 200 10/300 column (Cytiva) equilibrated in 10 mM Tris, 100 mM NaCl, pH 8.0. The peak corresponding to the monomeric protein was concentrated and stored at −80 °C until used.

The WT gorilla GII FV ectodomain was cloned into the pT350 vector and used as a template for generating the heparan-sulfate-binding mutants by site-directed mutagenesis. *Drosophila* S2 cells were stably transfected with the vectors, as described above. The ectodomains expression followed the same steps reported for the RBD production and after 6 days they were purified from the cell supernatants by affinity chromatography using a StrepTactin column (IBA) and SEC on a Superose 6 10/300 column (Cytiva) in 10 mM Tris, 100 mM NaCl, pH 8.0. The fractions within the peak corresponding to the trimeric ectodomain were concentrated in VivaSpin concentrators and stored at −80 °C until used.

### Crystallization

Crystallization trials were performed in 200 nanoliter sitting drops formed by mixing equal volumes of the protein and reservoir solution in the format of 96 Greiner plates, using a Mosquito robot. Crystal appearance and growth were monitored by a Rock-Imager at the Core Facility for Protein Crystallization at Institut Pasteur in Paris, France[48]. The native RBD$^D$ crystal used for data collection was grown in 0.1 M Tris pH 8.5, 3.5 M sodium formate (NaCOOH). For the derivative data, the RBD$^D$ crystal, grown in 0.1 M Tris pH 8.5, 3.25 M sodium formate, was soaked overnight in the same crystallization solution supplemented with 0.5 M sodium iodide and directly frozen using the mother liquor containing 33% ethylene glycol as cryo-buffer. The RBD$^G$ crystals

were obtained from a solution containing 0.2 M ammonium tartrate ((NH$_4$)$_2$ C$_4$H$_4$O$_6$) and 20% w/v PEG 3350.

## X-ray diffraction data collection and SFV RBD structure determination

The X-ray diffraction data were collected at the SOLEIL synchrotron source (Saint Aubin, France). The native data for RBD$^D$ and RBD$^G$ were collected at 100 K at the Proxima-1[49] beamline, at wavelength of 0.9786 Å, while the derivative (iodine-soaked) data for RBD$^D$ were collected at Proxima-2A, at wavelength of 1.907 Å. The beamlines are equipped with the Pilatus Eiger X 16 M and Eiger X 9 M detectors (Dectris), respectively.

We obtained trigonal crystals, space group 322$_1$ for the RBD$^D$ (2.57 Å), P3$_1$21 (later found to be P3$_2$21) for the derivative RBD$^D$ (3.2 Å), and hexagonal crystals for the RBD$^G$ protein (2.8 Å, space group P6$_1$). Diffraction data were processed using XDS[50] and scaled and merged with AIMLESS[51]. The high-resolution cut-off was based on the statistical indicator CC1/2[52]. Several applications from the CCP4 suite were used throughout processing[53]. The statistics are given in Table S1.

The phases were determined experimentally by single-wavelength anomalous diffraction. The AutoSol pipeline from the Phenix suite[54,55] was employed, using the anomalous data set, searching for iodine sites and specifying two NCS copies in the asymmetric unit (ASU). AutoSol reliably determined the substructure, composed of 20 iodine sites. The refined anomalous phases were internally used to phase the entire protein with the aid of density modification. The result of the process was a structure with a low R-factor; moreover, the density modified map showed a good contrast between the protein and the solvent and helical features clearly discernible. The initial assignment of the space group of the anomalous data was tentative, as the screw axis that is present in the cell allows for two alternatives (P3$_1$21 or P3$_2$21). The enantiomorph ambiguity was resolved after density modification with the anomalous phases and model building by looking at the map and its quality. AutoSol unambiguously selected the correct space group, which is P3$_2$21. The structure was further improved in Buccaneer[56] in 'experimental phases' mode, using the density modified map from AutoSol and the refined substructure from AutoSol. Finally, the BUCCANEER model was refined against the native data at 2.57 Å by iterative rounds of phenix.refine[54], BUSTER[57,58] and Coot[59], which was used throughout all model building and refinement to inspect and manually correct the model.

To solve the structure of the RBD$^G$, the RBD$^D$ structure was used as a search-model in Molecular Replacement in Phaser[60] from the Phenix suite. In this case, the ASU was found to contain two molecules, which were again refined using a combination of BUSTER and phenix.refine.

For both models, the 2Fo-Fc and Fo-Fc electron density difference maps were used to unambiguously identify the carbohydrate moieties and built them. For both models, the final stereochemistry was assessed by MolProbity (http://molprobity.biochem.duke.edu/)[61].

The final maps showed clear, interpretable electron density, except for a region comprising residues 420-426 precluding building on these 7 amino acids and indicating inherent flexibility of the region. The atomic models were refined to R$_{work}$/R$_{free}$ of 0.21/0.25 and 0.19/0.23, for the RBD$^D$ and RBD$^G$ crystals, respectively. The RBD$^D$ and RBD$^G$ models had 95.99% and 95.69% of residues within the favored region of Ramachandran plot, and 0.31% and zero outliers, respectively.

## SFV GII RBD fitting into PFV Env EM map

The RBD fitting was performed with the fit-in-map function in Chimera suite[29] using the RBD$^D$ model (PDB: 8AEZ) and the 8.8 Å EM map (EMD-4013) obtained for the PFV Env[13]. The map used for fitting was simulated from atoms at a 9 Å resolution and data above the map contour level 0.025, resulting in a correlation coefficient of 0.96. To prepare Fig. 4, a contour level of 0.014 was chosen to allow visualization of the density for most of the glycans.

## Cells, sequences, and production of Foamy virus viral vectors

Baby Hamster Kidney (BHK)−21 cells (ATCC-CLL-10) were cultured in DMEM-glutamax-5% fetal bovine serum (FBS) (PAA Laboratories). HT1080 cells (ECACC 85111505) were cultured in EMEM-10% FBS supplemented with 1x L-glutamine and 1x non-essential amino acids (NEAA). Human embryonic kidney 293T cells (CRL-3216) were cultured in DMEM-glutamax-10% FBS.

Foamy virus isolates were named according to the revised taxonomy[22] and short names were used for gorilla and chimpanzee strains[19]. The four-component FVV system (plasmids pcoPG, pcoPP, pcoPE, pcu2MD9-BGAL (a transfer plasmid encoding for β-galactosidase)) and the gorilla Env construct containing sequences from the zoonotic GI-D468 (JQ867465) and GII-K74 (JQ867464) env genes (EnvGI-SUGII) have been described[19,42]. Briefly, the genotype II Env construct we used (EnvGI-SUGII[19]) is comprised of the SU is from the GII-BAK74 genotype, and the LP and TM from the GI strain BAD468, the latter two being very conserved between GI and GII.

Mutations in the RBD predicted heparan sulfate-binding site (K342A/R343A and R356A/R369A) were introduced to this gorilla Env plasmid containing full-length GII Env. The Env ΔL2 and ΔL4 variants lacked residues 278-293 and 442-458, respectively, which were replaced by glycine linkers (GGGG for ΔL2 and GG for ΔL4). FVVs were produced by co-transfection of four plasmids (gag:env:pol:transgene β-galactosidase) at a ratio of 8:2:3:32. Three micrograms total DNA and 8 μl polyethyleneimine (JetPEI, #101-10N, Polyplus, Ozyme) were added to 0.5 × 10$^6$ HEK 293T cells seeded in 6-well plates. Supernatants were collected 48 h post transfection, clarified at 1500 × g for 10 min, and stored as single-use aliquots at −80 °C. Vector infectivity was determined by transducing BHK-21 cells with serial five-fold dilutions of vectors and detecting β-galactosidase expression after 72 h of culture at 37 °C. Plates were fixed with 0.5% glutaraldehyde in phosphate-buffered saline (PBS) for 10 min at room temperature, washed with PBS and stained with 150 μl X-gal solution containing 2 mM MgCl$_2$, 10 mM potassium ferricyanide, 10 mM potassium ferrocyanide and 0.8 mg/ml 5-bromo-4-chloro-3-indolyl-B-D-galactopyranoside in PBS for 3 h at 37 °C. Counting was done on a S6 Ultimate Image UV analyzer (CTL Europe, Bonn, Germany), with one blue cell defined as one infectious unit. Cell transduction by FVV is a surrogate for viral infectivity and FVV titers were expressed as infectious units/ml.

The yield of FVV particles was estimated by the quantification of particle-associated transgene RNA. FVV RNAs were extracted from raw cell supernatants with QIAamp Viral RNA Extraction Kit (Qiagen). RNAs were treated with DNA free kit (Life Technologies), retro-transcribed with Maxima H Minus Reverse Transcriptase (Thermo Fischer Scientific) using random primers (Thermo Fischer Scientific), according to manufacturer's instructions. qPCR was performed on cDNA using BGAL primers (BGAL_F 5′ AAACTCGC AAGCCGACTGAT 3′ and BGAL_R 5′ ATATCGCGGCTCAGTTCGAG 3′) with a 10 min denaturation step at 95 °C and 40 amplification cycles (15 s at 95 °C, 20 s at 60 °C and 30 s at 72 °C) carried out with an Eppendorf realplex2 Mastercycler (Eppendorf). A standard curve prepared with serial dilutions of pcu2MD9-BGAL plasmid was used to determine the copy number of FVVs. Results were expressed as vector particles/ml, considering that each particle carries 2 copies of the transgene.

## Prediction of RBD heparan-binding site and mutant design

The server ClusPro (https://cluspro.org/login.php) was used for identifying a potential heparin-binding site[31,62–64]. The server generated 13 models of a fully sulfated tetrasaccharide heparin fragment docked to the FV RBD and a list of atom-atom contacts between the heparin chain and the protein residues that was used to generate the plots on Fig. 5b.

## Env interactions with heparan sulfate assayed by binding to heparan-sulfate Sepharose

One-hundred micrograms of recombinant FV RBDs (wild-type, R356A/R369A, K342A/R343A) were injected at 1 ml/min onto a Heparin-Sepharose column (Cytiva) previously equilibrated with running buffer (10 mM Tris, 100 mM NaCl, pH 8.0). After washing, a linear gradient (from 0 to 50% over 30 min) of elution buffer (10 mM Tris, 2 M NaCl, pH 8.0) was applied.

## Env interactions with heparan sulfate on cells (in vitro): Env protein-binding assay

HT1080 and BHK-21 adherent cells were detached with Trypsin-EDTA and $5 \times 10^5$ cells were used per condition. Cell washing and staining steps were performed in PBS, 0.1% bovine serum albumin (BSA) at 4 °C. SFV Env ectodomains were added to the cell pellet for 1 h. Cells were washed twice, incubated with StrepMAB-Classic-HRP antibody that recognizes the strep tag at the C-terminus of the SFV Env ectodomain (7.5 µg/ml, IBA Lifesciences #2-1509-001) for 1 h, washed twice and incubated with the secondary antibody coupled to fluorophore AF488 (anti-HRP-AF488 (0.75 µg/ml, Jackson ImmunoResearch, #123-545-021)) for 30 min. Cells were washed and fixed in PBS, 2% PFA at room temperature for 10 min and kept at 4 °C until acquisition. A minimum of 25,000 cells were acquired on a Cyto-FLEX cytometer (Beckman Coulter). Data were analyzed using Kaluza software (Beckman Coulter). Viable single cells were selected by the sequential application of gates on FSC-A/SSC-A and SSC-A/SSC-H dot-plots (Fig. S13a). Cells labeled with the two secondary antibodies only were used as a reference. SFV Env binding was expressed as the ratio of mean fluorescence intensity (MFI) from the cells that were incubated with the recombinant ectodomains vs. untreated cells (Fig. S13b).

## Heparan sulfate removal and detection

Cells were treated with Trypsin-EDTA and $5 \times 10^5$ cells were labeled per condition. Cells were washed once with PBS, 0.1% BSA prior to incubation with 0.1 mIU/ml heparinase III from *Flavobacterium heparinum* (Sigma-Aldrich, #H8891) in 20 mM Tris, 0.1 mg/ml BSA and 4 mM $CaCl_2$, pH 7.45 for 15 min. at 37 °C. Heparan sulfate was detected by staining with F58-10E4 antibody (5 µg/ml, AmsBio, UK #370255-S) and anti-mouse IgM-AF488 antibodies (2 µg/ml, Invitrogen #A-21042). The neoantigen generated by HS removal (ΔHS) was detected with the F69-3G10 antibody (10 µg/ml, AmsBio #370260-S) and anti-mIgG-AF647 antibodies (4 µg/ml, Invitrogen #A-31571). Cell staining and washing were performed in PBS, 0.1% BSA at 4 °C. Incubation times were 60 and 30 min for primary and secondary antibodies, respectively. Cytometer acquisition, and data analysis were performed as described for Env binding (Fig. S13). Cells labeled with secondary antibodies only were used as a reference. Levels of HS and ΔHS staining were expressed as the ratio of MFI from labeled to unlabeled cells (Fig. S13c).

## FVVs-binding assay

HT1080 cells were incubated with FVV particles (1, 10, and 100 particles/cell) on ice for 1 h. Cells were washed three times with PBS to eliminate unbound FVVs and RNAs were extracted using RNeasy plus mini kit (Qiagen) according to manufacturer's protocol. RT was performed as described for FVVs RNA quantification. Bound FVV were quantified by qPCR of *bgal* gene as described for vector titration; cells were quantified by a qPCR amplifying the *hgapdh* gene with the following primers: hGAPDH_F 5' GGAGCGAGATCCCTCCAAAAT 3' and hGAPDH_R 5' GGCTGTTGTCATACTTCTCATGG 3'. The qPCR reaction conditions were the same as those used to amplify the *bgal* gene. Relative mRNA expression of *bgal* versus *hgapdh* was calculated using the −ΔΔCt method, and relative binding as $2^{-\Delta\Delta Ct}$.

## Statistics

The infectious titers, particle concentration, percentages of infectious particles and quantity of bound FVVs carrying WT and mutant Envs were compared using the two-way paired *t*-test.

## Reporting summary

Further information on research design is available in the Nature Portfolio Reporting Summary linked to this article.

## Data availability

The data generated in this study and related to the X-ray structures determined for the SFV GII RBD$^D$ and RBD$^G$ have been deposited to the RCSB protein databank under PDB accession codes 8AEZ and 8AIC, respectively. The AF2 models of FV RBDs have been deposited to the Model Archive database, with the following accession codes: gorilla (genotype II, accession code: ma-5hiw1), gorilla (genotype I; accession code: ma-sln9b), prototype Foamy virus (chimpanzee, genotype I; accession code: ma-ogxjm), Western chimpanzee (genotype I; accession code: ma-zilao), Central African chimpanzee (genotype II, accession code: ma-u3aws), African green monkey (accession code: ma-mf4i2), orangutan (accession code: ma-kae1t), macaque (accession code: ma-eolif), marmoset (accession code: ma-4q50y), bovine (accession code: ma-ad22f), equine (accession code: ma-iodkg), and feline (accession code: ma-ocsub). Source data are provided with this paper.

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

## Acknowledgements

This work was funded by the 'Agence Nationale de la Recherche' (ANR-10-LABX62-IBEID, Intra-Labex Grant, M.B.), the 'Programme de recherche transversal from Institut Pasteur' (PTR2020-353 ZOOFOAMENV, F.B.), and recurrent funding from Institut Pasteur (F.A.R, A.G.). The funding agencies had no role in the study design, generation of results, or writing of the manuscript. We thank the staff from the Utechs Cytometry & Biomarkers and Crystallography platform at the Institut Pasteur, the synchrotron source SOLEIL (Saint-Aubin, France) for granting access to the facility, and to the staff of Proxima 1 and Proxima 2A beamlines for their kindness and assistance during X-ray data collections. We are grateful to Jan Hellert, Pablo Guardado-Calvo and Philippe Afonso for the discussions and advice, with special thanks to Max Baker for reading the manuscript and English language corrections.

## Author contributions

M.B. and F.B. conceived and supervised the study. I.F. and L.T.D. carried out the bulk of the experiments. I.F. crystallized the RBDs with the help of A.H., collected the X-ray diffraction data, solved the structures with R.P. and carried out the binding studies with heparin Sepharose column. L.T.D. performed all the binding studies involving the cells. Y.C. produced and assessed Foamy virus vector particles in infection assays. D.B. provided ongoing technical assistance. The original manuscript draft was written by M.B., with the input from I.F., R.P., L.T.D. and Y.C., and the review and editing were done by M.B., I.F., L.T.D., F.B., and F.A.R. The funding was acquired by M.B., F.B., F.A.R., and A.G.

## Competing interests

The authors declare no competing interests.
