## [Peer Review File · Nature Communications]

Reviewer comments

Reviewer #1 (Remarks to the Author):

Backovic and colleagues present the crystal structure of the RBD of simian Foamy virus Env. FV Env RBD folds into two domains that are evolutionary distinct from gamma-retroviruses and lentiviruses. The structure was docked into a medium resolution cryoEM map of complete Env, providing a first clue of the trimeric arrangement of RBD and its position within trimeric Env. Mutagenesis was used to define four basic residues implicated in heparin sulfate binding, which localize to the lower domain of RBD. The role of the basic residues in attachment/entry was confirmed by recombinant infectious particles.

In summary, the presented work is of high technical quality and of interest to the virology community working on Foamy virus biology and/or developing Foamy virus vectors for gene therapy.

I have only a few comments that the authors need to address:

Fig. S8 AlphaFold modeling: Were all structures shown in S8 modelled ab initio or was the X-ray structure used as a template for the modeling?

Fig. S10: RBD fitting into the EM map indicates that the interface residues are mostly not conserved, especially within the bottom trimer interface comparing 3 SFV RBDs. Can the authors comment on this since the trimer interface would be expected to be conserved. Is this true for all 11 AF models? Please also indicate the interface residues making potential contacts at the interface in S9.

In order to strengthen the trimer model, the authors should validate their model by cysteine mutagenesis/disulfide bonds.

Reviewer #2 (Remarks to the Author):

This paper reports the X-ray structure of the receptor binding domain (RBD) of a simian Foamy Virus Env at 2.6 Å resolution, describing for the first time a new fold. Using a previous low resolution EM structure, they obtained a model for the organization of the RBDs within the trimeric Env that seems to indicate that the upper subdomain is important for stabilization of the full-length Env. They also showed that residues K342, R343, R359 and R369 in the lower subdomain play key roles in the interaction of the RBD and viral particles with heparan sulfate.

The paper is well written with numerous figures and supplemental information.

The authors make many comparisons to the HIV-1 Env, which are justified in some places but not necessarily needed in the abstract. Additionally maybe the authors could explain in more details why it is of interest to obtain the structure of the RBD of a FV. That justification/interest seems to be lacking at places.

Since the RBD crystal structure and AF model are so close, which is pretty amazing, one could wonder what is the AF model of the trimer including the TM ectodomain. Is there a reason why the authors did not include it.

Is there a particular reason the authors did not make a construct including the 4 mutations? It seems that such construct might be beneficial in trying to look for the potential proteinaceous receptor.

The hypothesis that the loops in the "upper domain" are responsible for trimer interactions and maintain the prefusion conformation is interesting and it would be nice to see if it could be tested with mutants lacking the loops. Although it appears that some were made in accompanying manuscript.

Minor comments;

- Line 53 - remove as in is 'as' an
- In results - could state how the protein was deglycosylated.
- Fig 5A - looks like the cartoon representation orientation is different from the surface to the left by looking at the stick residues.
- Maybe it is not that surprising that the fold of the various Env RBD protein is different since they belong to different retrovirus subfamilies...a mention of this could be added.

Reviewer #3 (Remarks to the Author):

The authors reported the crystal structure of a simian Foamy virus RBD, which would provide clues for the entry into host cells. The manuscript is well written and provides the structural and biochemical and also cellular evidences for the essential role of the RBD. The PDB validation reports show that the crystal structures are well determined and refined. There are some suggestions to the modification of the manuscript as the following:

- (1) In the abstract, 2.57 Anstrom should be used instead of 2.6 Anstrom, which should be consistent with that in the main text.
- (2) The authors stated that N8 likely plays an important structural role in all FV RBDs. It is better to compare the structures of RBDD and RBDG to see whether there are some local conformational changes around N8.
- (3) AlphaFold2 (AF2) should be used instead of AlphaFold (AF).
- (4) Is it possible to dock heparin into the crystal of RBD to provide the structural model to explain the molecular mechanism of how K342, R343, R356, R369 could play a crucial role in virus interaction with HS? Or did the authors try to get the co-crystal structure of RBD and heparin?
- (5) In line 421, 2Fo-Fc and Fo-Fc electron density maps should be used. These are difference maps.
- (6) In line 411, the enantiomorph ambiguity was not just resolved after density modification. Please check the crystallography textbook about the SAD theory. And the authors should check whether the phases were determined by SAD or SIRAS.
- (7) Based on the crystal structure, the RBD could be a trimer and how to get the monomer of RBD? What is the triggering factor for the monomer of RBD to be trimer?
- (8) When the authors state that the RBD fold is predicted to be conserved with the subfamily, it is better to superimpose all or some of the structures (shown in the backbone) to show a direct illustration of the conservation of the fold.

RESPONSE TO THE REVIEWERS' COMMENTS

Dear Reviewers,

Thank you very much on your valuable feedback, and remarks and suggestions on how to improve our manuscript. Please find the point-to-point answers, in blue, in the text below.

Please note that the **line numbers** we refer to in this letter correspond to those in the manuscript file (**Fernandez_manuscript_R.1_tracking.docx**) that has the "Tracking function" on, showing all the implemented changes. The line numbers in the PDF file are different.

Best wishes,

Marija Backovic, on behalf of all co-authors

REVIEWERS' COMMENTS

Reviewer #1 (Remarks to the Author):

Backovic and colleagues present the crystal structure of the RBD of simian Foamy virus Env. FV Env RBD folds into two domains that are evolutionary distinct from gamma-retroviruses and lentiviruses. The structure was docked into a medium resolution cryoEM map of complete Env, providing a first clue of the trimeric arrangement of RBD and its position within trimeric Env. Mutagenesis was used to define four basic residues implicated in heparin sulfate binding, which localize to the lower domain of RBD. The role of the basic residues in attachment/entry was confirmed by recombinant infectious particles.

In summary, the presented work is of high technical quality and of interest to the virology community working on Foamy virus biology and/or developing Foamy virus vectors for gene therapy.

I have only a few comments that the authors need to address:

- Fig. S8 AlphaFold modeling: Were all structures shown in S8 modelled *ab initio* or was the X-ray structure used as a template for the modeling?

All the structures were generated by *ab initio* AlphaFold 2 modelling. We have included this information in the manuscript (**lines 152-153**) and in the Figs S8 legend.

- Fig. S10: RBD fitting into the EM map indicates that the interface residues are mostly not conserved, especially within the bottom trimer interface comparing 3 SFV RBDs. Can the authors comment on this since the trimer interface would be expected to be conserved. Is this true for all 11 AF models? Please also indicate the interface residues making potential contacts at the interface in S9.

We thank the reviewer for this important remark. We analyzed the inter-RBD interfaces in our trimeric RBD model with ePISA and ProteinTools software and indicated some of the interface residues making potential contacts on new panels B and C on Fig. S10. We added the text describing these observations in **lines 195-197** of the Results section. However, those contacts residues must be taken

with caution because of several factors that limit the accuracy of our model: 1) the fitting of the RBD crystal structure was performed with a map obtained for a different virus (gorilla genotype II RBD vs PFV, which is chimpanzee genotype I Env), 2) the resolution of the map (9 Å) does not allow refining the position of side chains, 3) 6 residues could not be built into L3, and all the loops (L1-L4) have high B factors. In addition, AF2 assigned low pLTTD values to the residues in the loops of other FV RBDs, also precluding a reliable fitting of their loops into the EM map. We have added these pieces of information in the newly added paragraph explaining the "Limitations of our study", **lines 363-370** in the Discussion section.

We agree that conservation of the residues at the interface would be expected, but our experiments with the Env loop deletion mutants (new Fig. S11) indicate that despite poor conservations the L2 and L4 play important structural and / or functional roles (**lines 201-206**). One possible explanation is that maintaining the inter-protomer interface depends on main-chain interactions rather than side chains, or that the residues are not conserved but covary to preserve the interface. Unfortunately, there are not enough FV Env sequences to validate this point.

Regarding conservation of the loop residues, we highlighted our hypothesis that the RBDs within the trimeric Env are likely bound loosely, interacting through relatively small surfaces formed by the loops, whose sequences would therefore not have been under a strong selective pressure (**lines 311-318** of the Discussion), explaining their low conservation. The small BSA between the RBDs could also be an indication of potential loop dynamics at the top of the Env, akin to breathing of the HIV Env (**lines 318-319**). In that respect it is interesting that the lower RBD subdomain seems to be in contact with the TM part of the Env and contains a large hydrophobic core (Fig. S1), making it likely more rigid in comparison with the upper subdomain and loops that have a hollow space below, and could be conformationally flexible.

- In order to strengthen the trimer model, the authors should validate their model by cysteine mutagenesis/disulfide bonds.

We appreciate this excellent suggestion and agree that this would be the best strategy to prove that the loops maintain a closed pre-fusion Env conformation. Having an atomic resolution model of the trimeric Env would enable us to make educated guess and predict the pairs of residues at the distances and geometry conducive to formation of SS bonds. As mentioned above, our X-ray structure lacks 6 residues in L3, and our current trimeric RBD model does not provide sufficient accuracy for location of the loop residues, necessitating testing potentially a very large panel of mutants with no clear evidence that we would succeed in identifying a useful mutant in a reasonable amount of time. We have instead been streaming our efforts and resources towards getting atomic resolution structure of the pre-fusion Env trimer, which will be the proof of our model and will provide mechanistic insights (this is work in progress and is beyond the scope of this manuscript). In addition, it is possible that the RBDs in the FV Env undergo 'breathing' and are not fixed in place, as was described for HIV Env (PMCID: PMC6034635). We added text related to this possibility in Discussion section, **lines 284-287** and **318-319**.

As an alternative to the disulfide mutants, in this manuscript version, we provide the results on Env mutants with loop deletions. The data show that FVVs carrying Env with L2 and L4 deletions, have a modest reduction in binding to cells, but completely lose infectivity (new Fig. S11, **lines 201-206**). These results demonstrate that, even though not conserved in sequence, L2 and L4 play an important functional role in the Env context and are in agreement (while not being the proof) of the hypothesis that without the loops, the Env would spontaneously flip into inactive, post-fusion state. We mention the limitation of the loop deletion mutant studies in **lines 371-377**.

Reviewer #2 (Remarks to the Author):

This paper reports the X-ray structure of the receptor binding domain (RBD) of a simian Foamy Virus Env at 2.6 Å resolution, describing for the first time a new fold. Using a previous low resolution EM structure, they obtained a model for the organization of the RBDs within the trimeric Env that seems to indicate that the upper subdomain is important for stabilization of the full-length Env. They also showed that residues K342, R343, R359 and R369 in the lower subdomain play key roles in the interaction of the RBD and viral particles with heparan sulfate.

The paper is well written with numerous figures and supplemental information.

- The authors make many comparisons to the HIV-1 Env, which are justified in some places but not necessarily needed in the abstract.

We agree. We removed the sentence from the abstract.

- Additionally maybe the authors could explain in more details why it is of interest to obtain the structure of the RBD of a FV. That justification/interest seems to be lacking at places.

Thank you for this suggestion. We added text related to the importance and interest in this structure on lines 79-83.

- Since the RBD crystal structure and AF model are so close, which is pretty amazing, one could wonder what is the AF model of the trimer including the TM ectodomain. Is there a reason why the authors did not include it.

We were wondering the same. We tried predicting the FV Env ectodomain trimer, but the AF server we use at the Institute could not handle the size (an Env protomer is almost 1,000 residue long).

- Is there a particular reason the authors did not make a construct including the 4 mutations? It seems that such construct might be beneficial in trying to look for the potential proteinaceous receptor.

Thank you for this question. We did generate the RBD with 4 mutations, and as expected this RBD variant did not bind to the heparin column. The protein expression yields were low and since the results with the heparin column were confirmatory, and in the interest of time, we decided not to pursue the cell binding experiments. We agree that this RBD mutant will be the best choice for receptor fishing, and we plan to use it as such.

- The hypothesis that the loops in the “upper domain” are responsible for trimer interactions and maintain the prefusion conformation is interesting and it would be nice to see if it could be tested with mutants lacking the loops. Although it appears that some were made in accompanying manuscript.

We appreciate the suggestion. These data were originally included in the manuscript that we co-submitted with our collaborator Florence Buseyne (referred to as Dynesen *et al.* in our original manuscript). Because that manuscript will be submitted to another journal, we moved the data related to the gorilla GII Env loop deletions to our manuscript (Fig. S11 and lines 201-206). The Envs with deletions of L2 and L4 show a modest reduction in binding to cells, but completely lose infectivity). These results are consistent with the hypothesis that loops are important for maintaining of the Env in the fusion-competent, closed state, akin to the role of V1 and V2 loops in HIV Env (PMCID: PMC5135367). But the definitive identification of the contact residues will require an atomic

resolution structure of the entire FV Env trimer. We commented on this point in the Discussion subsection 'Limitations of the study' (lines 363-370)

Minor comments;

- Line 53 - remove as in is 'as' an

Corrected, thank you.

- In results - could state how the protein was deglycosylated.

We have included this information in the Results (line 94).

- Fig 5A - looks like the cartoon representation orientation is different from the surface to the left by looking at the stick residues.

Thank you for noticing. We have corrected the error.

- Maybe it is not that surprising that the fold of the various Env RBD protein is different since they belong to different retrovirus subfamilies...a mention of this could be added.

We modified the text on lines 271-272 to reflect this point clearly.

Reviewer #3 (Remarks to the Author):

The authors reported the crystal structure of a simian Foamy virus RBD, which would provide clues for the entry into host cells. The manuscript is well written and provides the structural and biochemical and also cellular evidences for the essential role of the RBD. The PDB validation reports show that the crystal structures are well determined and refined. There are some suggestions to the modification of the manuscript as the following:

- In the abstract, 2.57 Anstrogm should be used instead of 2.6 Anstrogm, which should be consistent with that in the main text.

Thank you. We took note and changed all the values to 2.57 Å

- The authors stated that N8 likely plays an important structural role in all FV RBDs. It is better to compare the structures of RBDD and RBDG to see whether are some local conformational changes around N8.

Thank you for this comment. The graphs with the buried surface area and presence of hydrogen bonds for each monosaccharide on Fig. S5 show that the two parameters are the same in the glycosylated and deglycosylated form, suggesting that the N8 conformation is unchanged in the two forms. To show more clearly that the polypeptide chain does not undergo any local conformational changes around N8, we magnified the relevant region and included this as a new panel C on Fig. S3.

- Alphafold2(AF2) used be used instead of AlphaFold (AF).

Thank you, we corrected this in the main text and in SI.

- Is it possible to dock heparin into the crystal of RBD to provide the structural model to explain the molecular mechanism of how K342, R343, R356, R369 could play a crucial role in virus interaction with HS? Or did the authors try to get the co-crystal structure of RBD and heparin?

Thank you for this remark. Docking HS using ClusPro program (Fig. 5A, **lines 547-550** in MM) is how we identified potential residues forming contacts with HS (K342, R343, R356 and R369). We modified the text, **lines 212-212**, to make it clear that docking was a part of the process. The 4 identified residues had the highest number of contacts with the docked HS molecules, which is why we picked them for further mutagenesis and functional studies, which confirmed that these residues are essential for RBD interactions with cells (Fig. 6C, D) and for infectivity (Fig. 6E, F).

There are several reasons why we did not try co-crystallization studies: 1) the functional data were convincing and demonstrated involvement of the residues in binding to HS on cells, 2) co-crystallizations with HS can be challenging because of heterogeneity of glycosaminoglycans (GAGs) in heparin mixtures that are commercially available, and because the GAGs can bind to the protein in multiple orientations (PMID: 17229412). All together we were not sure that additional information would be gained from a HS-RBD structure, relative to the time we would have had to invest.

- In line 421, 2Fo-Fc and Fo-Fc electron density maps should be used. These are difference maps.

We corrected this, thank you.

- In line 411, the enantiomorph ambiguity was not just resolved after density modification. Please check the crystallography textbook about the SAD theory. And the authors should check whether the phases were determined by SAD or SIRAS.

Thank you for this point! We added the information that the phases were determined by SAD (**line 478**). We understand the remark and agree that, in the context of SAD-phasing, density modification by itself does not resolve the enantiomorph ambiguity. In response to the remark, the sentence “The enantiomorph ambiguity was resolved after density modification with the anomalous phases and model building by looking at the map and its quality” has been removed from the text.

- Based on the crystal structure, the RBD could be a trimer and how to get the monomer of RBD? What is the triggering factor for the monomer of RBD to be trimer?

The recombinant RBD is a monomer in solution, and we did not observe any signs of RBD homotrimerization on SEC. Based on our fitting of the RBD structure into the EM map, the three RBDs come together at the top of the Env trimer, which is as all class I fusion proteins expressed as a trimer. Trimerization interfaces in these proteins are formed between different segments of protomers, with a central TM-subunit trimeric coiled coil that is their hallmark, and which was clearly observed in the PFV Env reconstruction (Effantin *et al.*). We think that the trimerization of the Env parts outside of RBD, namely TM subunit, is the primary trimerization driving force, which is also important for bringing the RBDs into the position at the top of the Env, where they can associate into a trimeric structure that is possibly kept by weak interactions between the RBDs. These RBD-RBD interactions may contribute to the stability of trimeric Env, but are not sufficient to drive trimerization of isolated RBDs in solution. We added a sentence to underscore this point in **lines 197-200**.

- When the authors state that the RBD fold is predicted to be conserved with the subfamily, it is better to superimpose all or some the structures (shown in the backbone) to show a direct illustration of the conservation of the fold.

We agree. We have added superposition of all the RBD models as panel B in Fig. S9.

Reviewer comments, second round

Reviewer #1 (Remarks to the Author):

The authors have responded to my previous questions. I was just wondering why the EM map shown in figure 4 shows much less details than the 9 A map described in the original publication? Model fitting is not described in the methods section. Why did the authors choose a lower cut-off for the map representation?

Reviewer #2 (Remarks to the Author):

The authors addressed most of the comments however it will be useful if the authors can add in discussion or results (as deemed appropriate) a sentence regarding "trying predicting the FV Env ectodomain trimer, but the AF server we use at the Institute could not handle the size (an Env protomer is almost 1,000 residue long)." as well as mention that they "generated the RBD with 4 mutations, with low expression yield, but that as expected this RBD variant did not bind to the heparin column...".

RESPONSE TO THE REVIEWERS' COMMENTS

Reviewer #1 (Remarks to the Author):

The authors have responded to my previous questions. I was just wondering why the EM map shown in figure 4 shows much less details than the 9 A map described in the original publication? Model fitting is not described in the methods section. Why did the authors choose a lower cut-off for the map representation?

Thank you for this comment. Figure 4 was prepared downloading the map from the EMDB and selecting a contour level that allowed a clear visualization of the membrane and showed density for most glycans (level ~0.014). The RBD structure was fitted with the 'Fit in map' tool from Chimera, which uses the average map value (level 0.025), which is the recommended contour level for map visualization in the EMDB Validation report. Therefore, the model docking has not been affected by the contour level chosen for preparing the figure. The model fitting was mentioned in the Results section, but we now added the information to Materials and Methods, to also clarify this point (lines 495 – 500).

Reviewer #2 (Remarks to the Author):

The authors addressed most of the comments however it will be useful if the authors can add in discussion or results (as deemed appropriate) a sentence regarding "trying predicting the FV Env ectodomain trimer, but the AF server we use at the Institute could not handle the size (an Env protomer is almost 1,000 residue long)." as well as mention that they "generated the RBD with 4 mutations, with low expression yield, but that as expected this RBD variant did not bind to the heparin column...".

Thank you. We have included the two pieces of information in lines 360-363 and 330-332, respectively.

N.B. We would like to add that we made a minor modification in the text and Figures 1 and S3 because we had overlooked the fact that site N10 had a single NAG built in molecule B of RBD^G. The electron density for site N10 was not observed in the RBD^D. The RBD^G crystals contained 2 molecules per asymmetric unit, where molecule A did not show density at N10, while molecule B did. Because we had overlooked the presence of N10 NAG in the latter molecule, we had reported that we did not observe density at N10 in either RBD^D or RBD^G. We have corrected that error in the main text (line 114) and in Fig. 1 and S3, where N10 is now indicated as being glycosylated. We also fixed a typo related to the sequence name for a chimpanzee SFV Env in Figure S6 legend.